# Cervical cancer care at a tertiary oncology facility in Uganda: Comparing daily practice with national treatment targets on cervical cancer control

**Marlieke de Fouw** [1]*, **Melissa W. M. Boere**[1], **Carolyn Nakisige**[2], **Mariam Nabwire**[2], **Jane Namugga** [2,3], **Israel Luutu**[4], **Jackson Orem**[5], **Jan M. M. van Lith**[1], **Jogchum J. Beltman**[1]

1 Department of Gynaecology, Leiden University Medical Centre, Leiden, The Netherlands, 2 Department of Gynaecological Oncology, Uganda Cancer Institute, Kampala, Uganda, 3 Mulago Specialised Women and Neonatal Hospital, Kampala, Uganda, 4 Department of Radiotherapy, Uganda Cancer Institute, Kampala, Uganda, 5 Uganda Cancer Institute, Kampala, Uganda

* marlieke@femalecancerfoundation.org

**Data Availability Statement:** All relevant data are within the manuscript.

## Abstract

### Objective

Treatment of cervical cancer patients in Uganda is hampered by late diagnosis due to the unavailability of timely screening and limited availability of advanced cancer care. This study evaluated the clinical presentation and management of cervical cancer patients presenting at the Uganda Cancer Institute (UCI) in Kampala, the tertiary oncology facility in Uganda with access to radiotherapy and reflected on daily clinical practice to identify priority areas for improving cervical cancer care in Uganda.

### Patients and methods

We retrospectively analyzed medical records of all cervical cancer patients presenting to UCI between January 2017 and March 2018 for sociodemographic characteristics and clinical variables with descriptive statistics. The clinical management of patients with early and advanced stage disease who initiated treatment at UCI was evaluated using the national targets formulated in the Uganda strategic plan for cervical cancer prevention and control.

### Results

Medical records of 583 patients were included, representing less than 10% of the annual estimated incidence in Uganda. The majority (86%) of patients presented with advanced stage of disease. More than half of patients never initiated (31%) or interrupted (30%) treatment. The national treatment targets for surgery (10%) and palliative care (25%) were achieved for eligible patients at UCI, however, the target for chemoradiotherapy (65%) was not met.

**Funding:** The author(s) received no specific funding for this work.

**Competing interests:** The authors have declared that no competing interests exist.

## Conclusion

Daily clinical practice differed from the ambitions formulated in the national treatment targets on cervical cancer control. While most women presented in advanced stage requiring chemoradiotherapy, the target was not met due to limited availability of radiotherapy. Although targets for surgery and palliative care were achieved at UCI facility level, they mask the unmet need of the majority of cervical cancer patients who never initiated or completed treatment. This demands for further expansion of oncological surgical capacity, chemotherapy and radiotherapy and warrants to focus on accessible prevention programs.

## Introduction

The treatment of cervical cancer patients in Uganda is hampered by late diagnosis due to low coverage of opportunistic preventive screening of women and the limited availability of advanced cancer care [1–3]. Cervical cancer is almost entirely preventable through primary and secondary prevention programs [4, 5]. Still, each year estimated 604.000 women are diagnosed with cervical cancer, over 85% occurs in low-and-middle-income countries (LMICs) [6]. The unequal distribution of cervical cancer burden between high-income countries (HICs) and LMICs is caused by multiple factors including the relatively high prevalence of human papillomavirus (HPV) infections and HIV infections in LMICs, the lack of well-organized prevention programs and the lack of accessible cancer care facilities [5, 7–10].

Uganda is one of the LMICs where cervical cancer is the leading cause of cancer death among women, and 80% of patients present with advanced stage of disease [1]. The Uganda Cancer Institute (UCI) in Kampala is the only oncology facility in Uganda where all treatment modalities for cervical cancer (surgery, radiation and chemotherapy) are provided. Access to radiotherapy, however, is limited. At the time of this study two cobalt machines need to cover all oncology patients.

In 2010 the Ministry of Health (MoH) in Uganda introduced its first strategic plan for national
cervical cancer prevention and control [11]. The strategic plan focused on several priority areas, including the treatment of cervical cancer patients, and defined specific targets for patient management to be achieved by 2015 (Box 1).

### Box 1. National treatment targets for cervical cancer control

1. 10 percent of eligible cervical cancer patients will be provided surgical treatment for invasive cervical cancer;

2. 65 percent of eligible women with cervical cancer will be provided with radiotherapy and chemotherapy services;

3. 25 percent of eligible cervical cancer patients will be provided palliative care services for improved quality of life.

To our knowledge, the treatment targets of Uganda have not yet been evaluated and studies reviewing the referral pathways and treatment of women with invasive cervical cancer in Sub-Saharan Africa are few [12–18]. This study evaluates the initial presentation, referral pathway and clinical management of patients presenting with cervical cancer to UCI and compares the clinical management at facility level with the national treatment targets. We reflect on cervical cancer care in daily clinical practice and aim to identify priority areas for improving cervical cancer care in Uganda.

## Materials and methods

### Study setting and participants

The present study isa retrospective analysis of medical records of all women presenting with biopsy confirmed cervical cancer between January 2017 and March 2018 at UCI, a tertiary oncology facility in Kampala, Uganda. Consultations, surgery and chemotherapy are provided free of charge, radiation for nominal fees. In March 2016 the single cobalt-60 radiation machine at UCI which facilitated both external beam radiotherapy and brachytherapy broke down. Patients had to move to neighboring countries like Kenya for radiation or were treated with an alternative protocol such as neo-adjuvant chemotherapy. In December 2017 a new cobalt machine was installed. In order to capture the effect of the re-introduction of radiotherapy patients who presented before and after instalment of the new cobalt machine were included.

At initial presentation to the gynae-oncology clinic at UCI all patients were registered in an Excel file with their referral diagnosis. Patients were included when the suspected cervical cancer diagnosis was confirmed with histopathology. Exclusion criteria were diagnosis other than cervical cancer, patients with cervical intraepithelial neoplasia or missing medical records.

### Data collection

One investigator [MB] and one trained research assistant and nurse extracted socio-demographic characteristics, reproductive and gynecological history, stage of disease and symptoms at presentation, diagnostic work-up, clinical management and follow-up data from medical records using a standardized data extraction sheet. The data were extracted between 17 May and 27 July 2018, anonymized and entered in Excel. The Excel file was imported into SPSS for statistical analysis.

The clinical performance status of patients was recorded using the Eastern Cooperative Oncology Group (ECOG) status [19]. The stage of disease was registered according to the 2009 International Federation of Gynaecology and Obstetrics (FIGO) staging system [20]. Currently the FIGO 2018 system is used, however, at the time of the study it was not yet used in clinical practice.

The recommended treatment for early stage cervical cancer (FIGO 2009 stage 1A-2A) is primary surgical treatment with curative intent. In case of contraindications for surgery, radiotherapy can be considered. For the advanced stages of disease (FIGO 2009 stage 2B-4A) a combination of chemotherapy and radiation with curative intent is recommended. For stage 4B the management consists of supportive palliative care.

### Statistical analysis

We analysed the following domains using descriptive statistics: sociodemographic and clinical characteristics at initial presentation, referral pathway and diagnostic work-up, and clinical

management compared to Ugandan national treatment targets (Box 1). The Chi-square test of independence was used for categorical sociodemographic and clinical variables between patients with early and advanced stage of cervical cancer, and the independent t-test for continuous variables. Statistical significance was set at p-value <0.05. For the evaluation of the national treatment targets patients were stratified in two groups: early stage (FIGO 2009 stage 1A-1B) and advanced stage (FIGO 2009 stage 2A-4B) cervical cancer. Patients with FIGO stage 2A were stratified as advanced stage because stage 2A was treated with a combination of chemotherapy and radiotherapy instead of primary surgical treatment as recommended in the FIGO international treatment guidelines. Additionally, patients were stratified according to date of presentation before (January to November 2017) and after (December 2017 to March 2018) instalment of the new cobalt radiation machine. In case of missing data for sociodemographic variables and clinical parameters, we calculated proportions using the total number of women for which the specific variable was recorded as denominator. All analyses were performed using IBM SPSS Statistics 25.

### Ethical considerations

Ethical approval was obtained from the Uganda Cancer Institute Research Ethics Committee with number 05/2018. The study was registered at the Uganda National Council on Science and Technology in May 2018 with number HS 2404. All data was collected retrospectively, anonymized, and recorded in password protected files on a secured server.

## Results

### Patient characteristics

Between 1st January 2017 and 31st March 2018 664 patients were registered at UCI with cervical cancer. In total 583 medical records (88%) could be retrieved and were included in the analysis. Table 1 presents the socio-demographic and reproductive health related patient characteristics. Mean age at presentation was 50.6 years (SD 13.0, range 20–90). Patients presenting in advanced stage were significantly older compared to patients in early stage of disease. Most patients (n = 287, 49%) resided in Central Uganda where UCI is located. About a quarter (n = 160, 27%) of the patients were known HIV-positive, of whom the majority (n = 151, 94%) used anti-retroviral treatment (ARVs) for a median duration of 7 years (IQR 4–10 years). HIV-status was not documented for another quarter (n = 141, 24%) of patients. HIV-positive patients presented on average 8.5 years younger than HIV-negative patients.

### Initial presentation

Fig 1 demonstrates the frequency of presentation specified per FIGO stage and HIV status. The majority of both HIV-positive (n = 126, 85%) and HIV-negative patients (n = 231, 86%) presented with advanced stage of disease.

Almost all patients (n = 564, 97%) reported symptoms on presentation, with abnormal vaginal bleeding (n = 453, 80%), pain (n = 408, 72%) and abnormal vaginal discharge (n = 384, 68%) as most common symptoms. Other regularly reported symptoms were poor appetite and difficulty with micturition or dysuria. Most patients (n = 420, 81%) presented in fair or good condition with ECOG performance status 0 or 1.

Previous screening for cervical cancer was reported by 404 patients (69%) of whom 9 had been treated with cryotherapy, loop electrosurgical excision procedure (LEEP) or unspecified treatment. The screening result and time duration since screening was not documented.

**Table 1. Patient characteristics, presented for the total population and separately for patients with early and advanced stage disease classified with the FIGO 2009 staging system.** The proportions are calculated over the total number of patients for which the variable is recorded, the proportion of missing values is calculated over the total number of patients.

| Patient characteristics | Total population | | Early stage (FIGO 1A-2A) | | Advanced stage (FIGO 2B-4B) | | p-value |
|---|---|---|---|---|---|---|---|
| | Number | Proportion (%) | Number | Proportion (%) | Number | Proportion (%) | |
| **Number of patients** | 583 | 100 | | | | | |
| • FIGO stage reported | 535 | 91.8 | 74 | 13.8 | 461 | 86.2 | |
| **Age** | 50.6 (13.0) | | 47.4 (14.1) | | | | **0.029** |
| • Mean (SD), years | | | | | 50.9 (12.6) | | |
| **Marital status** | | | | | | | 0.292 |
| • Married | 269 | 55.7 | 39 | 65.0 | 208 | 54.3 | |
| • Widowed | 136 | 28.2 | 11 | 18.3 | 112 | 29.2 | |
| • Single/separated | 77 | 15.9 | 10 | 16.7 | 63 | 16.4 | |
| • Unknown | 99 | 17.0 | 14 | 18.2 | 78 | 16.9 | |
| **Occupation** | | | | | | | 0.099 |
| • Peasant | 419 | 75.6 | 42 | 63.3 | 340 | 76.9 | |
| • Other | 135 | 24.6 | 24 | 36.7 | 102 | 23.1 | |
| • Unknown | 28 | 5.0 | 8 | 10.8 | 18 | 4.4 | |
| **District of residence** | | | | | | | 0.662 |
| • Central | 287 | 49.5 | 39 | 53.4 | 226 | 49.2 | |
| • East | 119 | 20.5 | 16 | 21.9 | 91 | 19.8 | |
| • West | 120 | 20.7 | 11 | 15.1 | 101 | 22.0 | |
| • North | 50 | 8.6 | 7 | 9.6 | 38 | 8.3 | |
| • Outside Uganda | 4 | 0.7 | 0 | 0 | 3 | 0.7 | |
| • Unknown | 3 | 0.5 | 1 | 1.4 | 2 | 0.4 | |
| **Menopausal status** | | | | | | | 0.252 |
| • Premenopausal | 244 | 45.0 | 36 | 52.2 | 193 | 41.9 | |
| • Postmenopausal | 298 | 55.0 | 33 | 47.8 | 238 | 51.6 | |
| • Unknown | 41 | 7.0 | 5 | 6.8 | 30 | 6.5 | |
| **Parity** | | | | | | | 0.365 |
| • Median (range), years | 6 (0–15) | | 5 (0–12) | | 6 (0–15) | | |
| • IQR 25–75 | 4–8 | | 4–8 | | 4–8 | | |
| • Unknown | 19 | 3.3 | 2 | 2.7 | 8 | 1.7 | |
| **Age at 1st sexual intercourse** | | | | | | | 0.926 |
| • Median (range) in yrs | 17 (9–28) | | 17 (13–28) | | 17 (9–28) | | |
| • IQR 25–75 | | | | | | | |
| • Unknown | 15–19 | 21.6 | 16–19 | 20.3 | 15–19 | 19.7 | |
| | 126 | | 15 | | 91 | | |
| **No. of sexual partners** | | | | | | | 0.786 |
| • 1 | 190 | 40.6 | 22 | 37.3 | 159 | 42.0 | |
| • 2–3 | 223 | 47.6 | 30 | 50.8 | 176 | 46.4 | |
| • 4 or more | 55 | 11.8 | 7 | 11.9 | 44 | 11.6 | |
| • Unknown | 115 | 19.7 | 15 | 20.3 | 82 | 17.8 | |
| **History of STI** | | | | | | | 0.416 |
| • Yes | 52 | 12.4 | 4 | 8.9 | 46 | 13.2 | |
| • No | 366 | 87.6 | 41 | 91.1 | 303 | 86.8 | |
| • Unknown | 165 | 28.3 | 29 | 39.2 | 112 | 24.3 | |

(*Continued*)

**Table 1.** (Continued)

| Patient characteristics | Total population | | Early stage (FIGO 1A-2A) | | Advanced stage (FIGO 2B-4B) | | p-value |
|---|---|---|---|---|---|---|---|
| | Number | Proportion (%) | Number | Proportion (%) | Number | Proportion (%) | |
| **HIV-status** | | | | | | | 0.912 |
| • Positive | 159 | 27.3 | 22 | 29.7 | 126 | 27.3 | |
| ○ CD4 <200 | 6 | 3.8 | 0 | 0 | 6 | 4.8 | |
| ○CD4 200–500 | 14 | 8.8 | 1 | 4.5 | 13 | 10.3 | |
| ○CD4 >500 | 18 | 11.3 | 3 | 13.6 | 13 | 10.3 | |
| ○CD4 missing | 121 | 76.1 | 18 | 81.8 | 94 | 74.6 | |
| • Negative | 282 | 48.4 | 36 | 48.6 | 231 | 50.1 | |
| • Unknown | 142 | 24.4 | 16 | 21.6 | 104 | 22.6 | |
| **ARVs in HIV-positive** | | | | | | | 0.707 |
| • Yes | 151 | 95.0 | 21 | 95.5 | 119 | 94.4 | |
| ○Median duration (range) in years | 7 (0–27) | | 9 (1–17) | | 9 (0–27) | | |
| ○IQR 25–75 | 4–10 | | 9–9 | | 9–9 | | |
| • No | 3 | 1.9 | 0 | 0 | 3 | 2.4 | |
| • Unknown | 5 | 3.1 | 1 | 4.5 | 4 | 3.2 | |

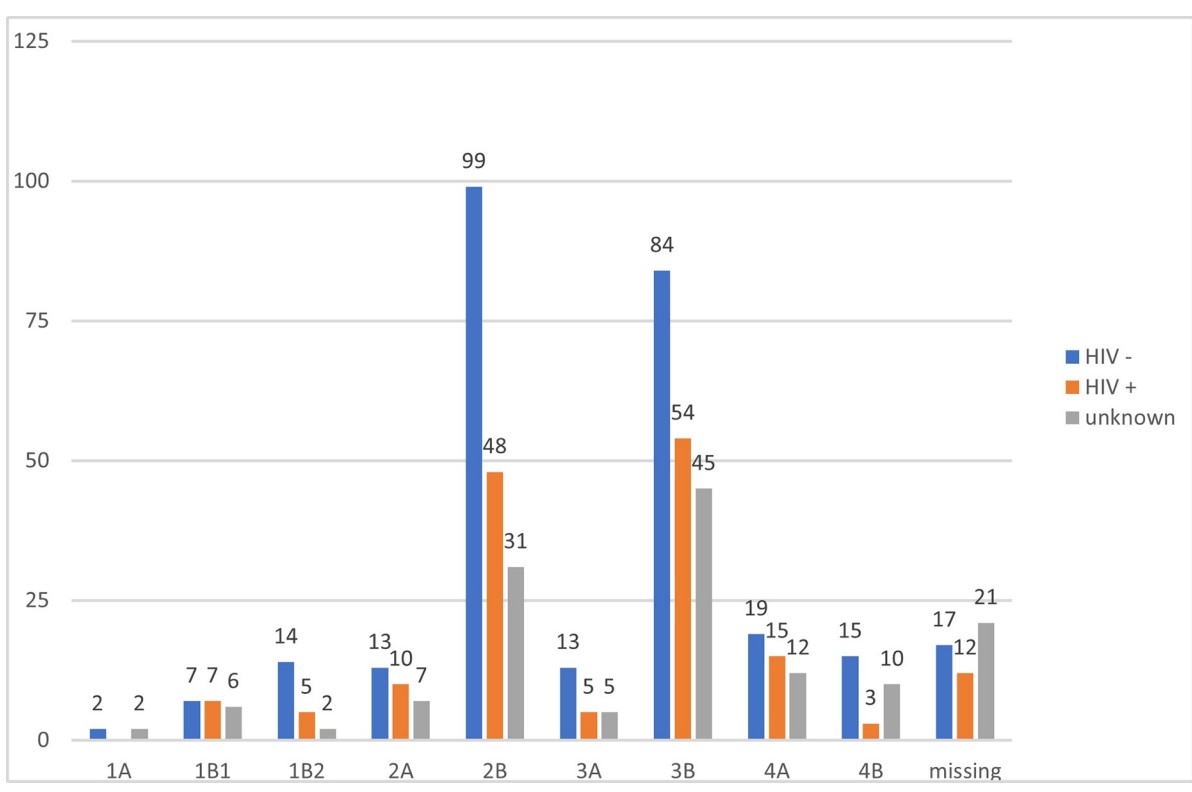

**Fig 1. Number of patients presenting between January 2017 and March 2018 with cervical cancer at UCI, specified per FIGO 2009 stage and HIV status.**

## Referral pathway

Three different referral pathways were identified: via district hospitals (n = 500, 86%), via screening programs (n = 34, 6%) and self-referral (n = 17, 3%). For the remaining patients referral data was not documented. The majority of patients (n = 26, 77%) referred by screening programs presented with advanced stage disease. A group of 36 patients (6%) presented with recurrent disease after previous treatment at UCI, of which 28 patients presented with advanced stage disease, for 6 patients the FIGO stage was not recorded.

## Diagnostic work-up

Before referral to UCI a biopsy was taken in 489 (84%) patients, with a known result in almost all patients (99%, n = 484), and 121 (21%) patients were examined by ultrasound.

In terms of management, analgesics were prescribed for 88 (15%) patients and 100 (17%) patients received other treatment like antibiotics. Simple hysterectomy was already performed in 73 (13%) patients in hospitals prior to referral to UCI for cervical cancer. Patients who underwent simple hysterectomy were symptomatic (n = 72, 99%), however, the medical records did not specify if simple hysterectomy was performed because cervical cancer was suspected or due to other indications like myoma. The pathology result showed squamous cell carcinoma in 65 patients and adenocarcinoma in 5 patients. In 3 patients the pathology result was not recorded in the file.

At UCI staging was based on clinical staging by pelvic examination. Additional investigations to complete staging of disease were ultrasound (n = 458, 79%), chest X-ray (n = 158, 27%) and MRI or CT-scan (n = 43, 7%). Pathology reports of biopsies and hysterectomy were available for 563 (96%) of patients: 72 from UCI, 491 from referring hospitals.

Table 2 illustrates the time between onset of symptoms, referral, presentation and start of treatment at UCI. No difference was observed between the patient groups with early and advanced stage disease in time between referral and presentation, and time between presentation and start of treatment.

Clinicians noted delays in the diagnostic work-up in 196 medical records, the most common reasons were the unavailability of the radiotherapy machine (n = 86, 44%), patient delay (n = 38, 19%), financial constraints (n = 22, 11%) and poor clinical condition (n = 21, 11%).

## Clinical management

During the study period 401 patients (69%) initiated treatment, of whom 160 patients (27%) completed treatment before April 2018, and 68 (12%) were still undergoing treatment (Fig 2). The remaining 173 patients (30%) interrupted treatment and did not attend their planned appointments. Another group of 182 patients (31%) never initiated treatment at UCI, more than 90% of these women had advanced stage disease.

**Table 2. Time between onset of symptoms, referral, first presentation and start of treatment at UCI (median, interquartile range IQR 25–75%), stratified by stage of disease.**

| Time period | Early stage | | Advanced stage | |
|---|---|---|---|---|
| | (FIGO 1A-2A) | | (FIGO 2B-4B) | |
| | *Number of patients* | *Median (IQR 25–75%)* | *Number of patients* | *Median (IQR 25–75%)* |
| Onset symptoms to presentation at UCI *(months)* | 49 | 5.0 (3.0–10.5) | 339 | 6.0 (3.0–12.0) |
| Date of referral to presentation at UCI *(days)* | 37 | 12.0 (4.0–25.5) | 212 | 8.0 (3.0–18.8) |
| Presentation at UCI to start treatment *(days)* | 54 | 24.0 (11.8–45.5) | 287 | 21.0 (11.0–38.0) |

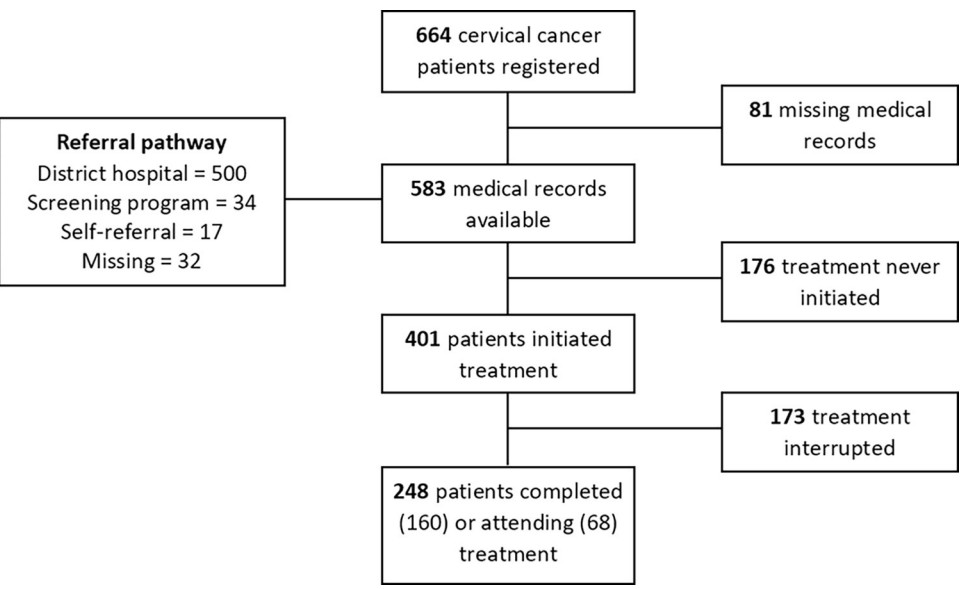

**Fig 2. Flowchart illustrating patient flow from initial registration at UCI to attending treatment.**

## National treatment targets

Figs 3 and 4 demonstrate that the national treatment targets for surgery (10% of eligible patients) and palliative care (25% of eligible patients) were met in both periods before and after instalment of the radiotherapy machine. The proportion of eligible patients who received surgical treatment was 35% and 50% respectively, 12 patients were treated with radical hysterectomy with pelvic lymphadenectomy, two patients were treated with simple hysterectomy.

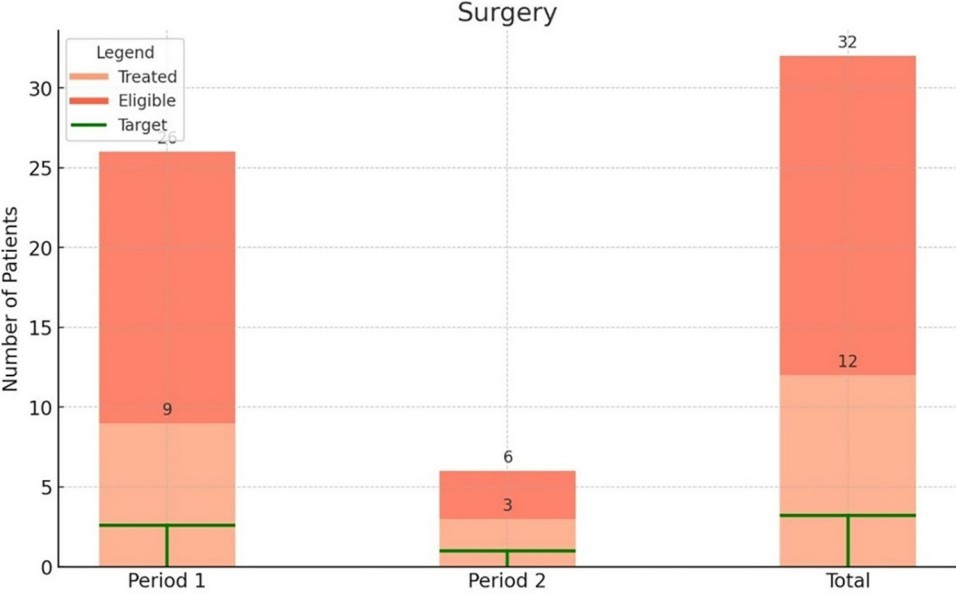

**Fig 3. Number of patients eligible for treatment (3 –surgery, 4 –palliative care, 5 –chemoradiation) and the number of patients that initiated the respective treatment at UCI, specified in period 1 before instalment of the radiotherapy machine and period 2 after instalment.** The target refers to the national treatment target as formulated in the Strategic Plan on cervical cancer control.

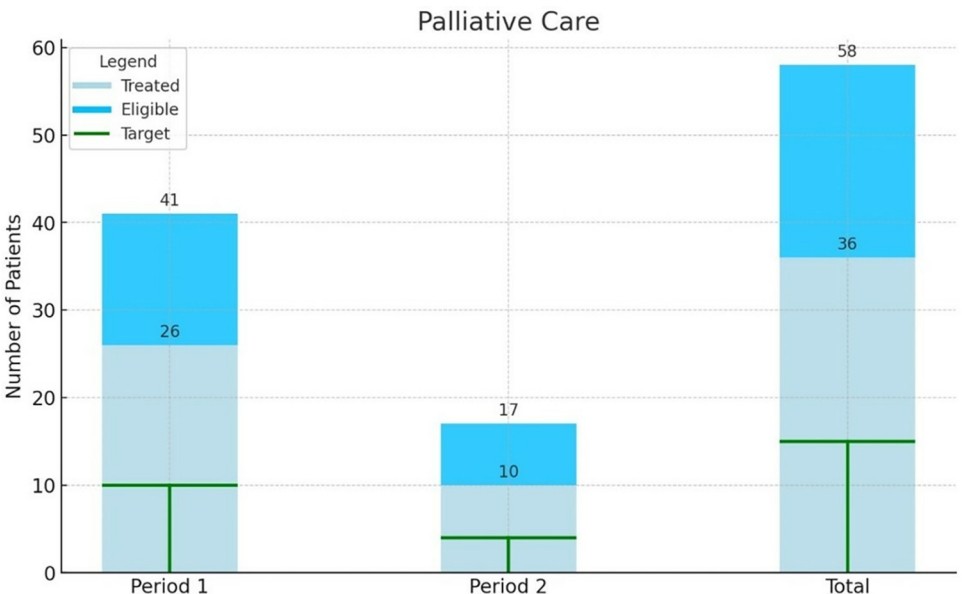

**Fig 4. Number of patients eligible for treatment (3 –surgery, 4 –palliative care, 5 –chemoradiation) and the number of patients that initiated the respective treatment at UCI, specified in period 1 before instalment of the radiotherapy machine and period 2 after instalment.** The target refers to the national treatment target as formulated in the Strategic Plan on cervical cancer control.

The proportion of eligible patients who received palliative care was comparable in the period before and after instalment, with 63% and 59% respectively.

The national treatment target of chemoradiotherapy (65% of eligible patients) was not met in either period (Fig 5). From all eligible patients 19% and 45% received chemoradiotherapy

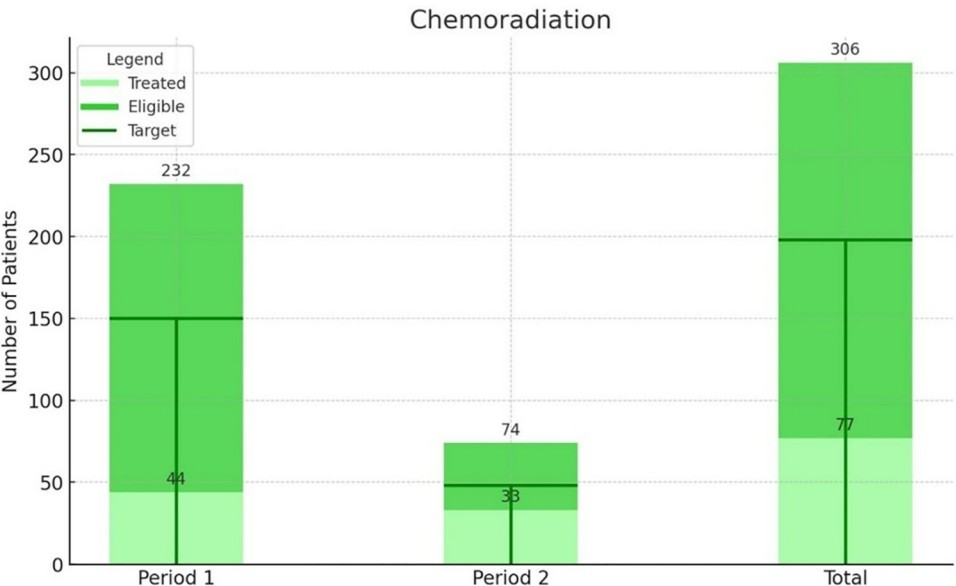

**Fig 5. Number of patients eligible for treatment (3 –surgery, 4 –palliative care, 5 –chemoradiation) and the number of patients that initiated the respective treatment at UCI, specified in period 1 before instalment of the radiotherapy machine and period 2 after instalment.** The target refers to the national treatment target as formulated in the Strategic Plan on cervical cancer control.

before and after instalment of the radiotherapy machine respectively. Patients were treated with alternative schedules, this included chemotherapy in 147 patients (63.4%), external beam radiotherapy (EBRT) in 12 patients (5.2%), high dose rate brachytherapy (HDR) in 18 patients (7.6%), and a combination of EBRT and HDR in 23 patients (9.9%) before instalment of the cobalt machine. After the reintroduction of radiotherapy at UCI this shifted to 6 patients (8.1%) receiving only chemotherapy, 12 patients (16.2%) EBRT, 3 patients (4.1%) HDR and 11 patients (14.9%) a combination of EBRT and HDR.

Patients in need of palliative care received chemotherapy in 26 cases (63.4%) in the period before the availability of radiotherapy, and in 10 cases (58.8%) after reintroduction of radio-therapy, besides other supportive care like analgesics and referral to other palliative care ser-vice providers like Hospice Uganda.

### Follow-up

The median follow-up duration for patients who completed treatment was 1 month (range 0 to 11 months). Patients who did not attend follow-up had a similar proportion of advanced stage of disease (n = 147 out of 166 with known FIGO stage, 89%) and similar ECOG status (n = 116, 75% ECOG 0 or 1) as the patients defaulting before initiating treatment and as the overall study population.

## Discussion

Our study evaluated the patient characteristics, referral pathway and clinical management of patients with cervical cancer presenting at UCI and is one of the first studies on tertiary care management of cervical cancer patients in Uganda. The majority of patients (86%) presented with advanced stage of disease, and more than half of the patients (60%) never initiated treat-ment or were lost to follow-up during treatment. Comparing daily clinical practice with the national treatment targets for cervical cancer control, we found that the targets for surgery and palliative care were achieved at facility level, however, the target for chemoradiotherapy was not met.

### Clinical presentation

Less than 10% of the annually estimated new cervical cancer patients in Uganda (6413 in 2018) presented to UCI [21]. This indicates that most patients in Uganda never reach UCI and prob-ably never receive any treatment for cervical cancer. Half of the registered patients originated from Central Uganda where UCI is located, suggesting that accessibility of tertiary care is even more limited for patients from other regions.

As reported in other studies from LMICs, the vast majority of women presented with advanced stage of disease [6, 7, 17, 22]. This correlates with our finding that almost all patients had symptoms at time of presentation. The large proportion of women with advanced stage of diseases illustrates the lack of a national organized screening program to identify and treat women with precancerous lesions and refer in early stage before onset of symptoms.

In our study population the HIV prevalence was 27.3%, which is high compared to the HIV prevalence (6.5–7.5%) in the general population of Uganda [23]. Combined with our finding that HIV-positive patients presented on average 8.5 years younger than HIV-negative patients, this supports the hypothesis that HIV-positive patients develop lesions at a younger age and progress more quickly to invasive cervical cancer than patients without HIV [24]. The dual burden of HIV infection and cancer in Uganda was also illustrated by a study conducted at UCI that reported a higher HIV prevalence and presentation at younger age of HIV-positive cancer patients compared to HIV-negative patients [25]. Interestingly, the proportion of HIV-

positive patients in our cohort was similar between patients with early and advanced stage of disease. HIV-positive patients might have better access to care and referral than HIV-negative patients and therefore present relatively more frequent to UCI than HIV-negative patients. A study conducted at UCI between 2003 and 2010 found that HIV-positive cervical cancer patients were less likely to present at advanced stage than HIV-negative patients [26]. Therefore, referral and access to tertiary cancer care are important factors to improve the outcome for cervical cancer patients.

### Diagnostic work-up and referral pathway

Patients presented themselves six months after the onset of symptoms at UCI. These findings are comparable with data from the tertiary Queen Elizabeth Central Hospital in Malawi and a study conducted between 2003 and 2010 at UCI [22, 26]. In the period between onset of symptoms and referral, patients often consulted different local clinics or health care providers. Although little data on referral pathways was available, this suggests an inadequately working referral system, insufficient knowledge about cervical cancer among referring health care providers and logistical constraints. Mwaka et al. conducted in-depth interviews in two district hospitals in Northern Uganda and found women do not seek care when symptoms are mild [7]. Repeatedly failed recognition of cervical cancer diagnosis by consulted health care providers further delayed referral.

A minority of patients was referred via screening programs. Nearly 70% of the patients reported to have participated in cervical cancer screening, while the lifetime screening rate in Uganda is reported between 4.8% and 30% [1]. The stark difference could be due to socially desirable answering by patients. It is unlikely that women in our study had considerably better access to the healthcare system and screening than the general population, since they presented with advanced stage of disease. Only 2.2% of women that attended screening reported subsequent treatment with cryotherapy and/or LEEP, indicating that adequate treatment in screening programs is often not available or not provided due to other reasons. Our findings illustrate the need for a screen-and-treat program with national coverage and efficient referral pathways in case of suspected cancer.

Before referral to UCI, one out of 8 patients underwent simple hysterectomy at district hospitals. In a few cases the preoperative diagnosis was reported to be myomas, in most cases the indication for surgery was not documented. Due to suboptimal surgery this led to inadequate staging once patients were referred to UCI for adjuvant therapy. Availability of histopathology and experience in clinical staging of cervical cancer in designated district hospitals could improve preoperative diagnosis, staging, and referrals to tertiary oncology facilities.

Remarkably, HIV status was unknown for 24% of patients at the time of presentation at UCI and HIV test results were not recorded in their medical records. It is unclear whether HIV testing was not offered, not recorded, or whether patients refused testing. This limits the understanding of clinical presentation and effectiveness of management in HIV-positive patients, while this group requires specific management.

### National strategic plan for cervical cancer control

UCI achieved the national target of surgical treatment for 10% of eligible patients at its facility. However, achieving the target still translates to inadequate treatment for 90% of patients with early stage of disease. Patients who would have good survival outcomes when receiving adequate surgical treatment. This underlines the importance of investing in oncological surgical capacity and infrastructure.

In advanced stage of disease, radiotherapy is an essential element of treatment [20]. While most patients presented with advanced stage of disease, Sub-Saharan Africa is the least developed region with respect to radiotherapy availability. Less than half of African countries have external beam radiotherapy and vaginal brachytherapy available [27–29]. Despite an increase in the number of radiotherapy facilities, no country had a capacity that matched the estimated need, and frequently the machines are old and poorly maintained.

Our study showed the consequences of this fragile system in Uganda. The only available radiotherapy machine at the time broke down, and even after instalment of another cobalt machine, the number of patients eligible for radiotherapy far exceeded its capacity. Even though the proportion of patients receiving chemoradiation more than doubled from 19% to 45%, still more than half of the eligible patients did not receive radiotherapy which negatively influences their symptom control and prognosis. Furthermore, the majority of patients presenting at UCI were eligible for chemoradiation, and therefore the lack of radiotherapy facilities affected most cervical cancer patients. Additionally, the waiting list is stretched even more by other cancer patients in need of radiotherapy [30]. One out of three cervical cancer patients never initiated treatment at UCI while 90% of this group had advanced stage of disease. It is likely that the limited availability of radiotherapy and the waiting list for treatment influenced their decision. To meet the recommended capacity for oncological care 45 radiotherapy units should have been available by 2020 in Uganda, and the demand is expected to increase to 66 units by 2030 [27–29].

For patients in need of palliative care, we found that around 40% did not receive any palliative care, while around half of the patients received palliative chemotherapy. This figure is consistent with the findings of Low et al. about end-of-life palliative care in Uganda, which showed that chemotherapy use in the last 30 days of life in Uganda is among the highest in the world [31, 32]. The authors emphasized the importance of developing evidence-based guidelines for end-of-life chemotherapy specific to sub-Saharan Africa, because the treatment has no demonstrated survival benefit.

## Strengths and limitations

The present study provides a comprehensive overview of the presentation and actual management of cervical cancer patients at UCI, and is one of the few studies reporting on tertiary cervical cancer care in LMICs. Moreover, the study captures the impact of access to radiotherapy for cervical cancer patients, which is severely limited in LMICs and has devastating clinical implications.

Including all cervical patients presenting over a period of more than a year mitigates selection bias and therefore can be considered as a representative presentation of daily clinical practice. On the other hand, for 12% of patients (n = 81), medical records could not be retrieved. The missing files were from patients presenting in different months, from different age groups and different districts, therefore selection bias is unlikely.

We aimed to evaluate clinical management, however, more than half of the patients never initiated treatment and could therefore not be included in the analysis. Therefore, the proportions of patients receiving care at UCI overestimates the actual need for treatment. Furthermore, the targets outlined in the national strategic plan evaluate clinical management at national level. Considering the small proportion of cervical cancer patients that reached UCI combined with the lack of other oncology facilities in Uganda, we concluded that the targets were not achieved at national level.

Moreover, the targets set by MoH do not aim for optimal treatment of all patients. Meeting the targets is a good first step in cervical cancer care in Uganda, but more is needed to make

quality care available for all patients affected by cervical cancer. In the updated national strategic plan published recently, no treatment targets for tertiary care are formulated and there is more emphasis on building infrastructure and capacity [33].

## Conclusion

This is one of the first studies evaluating the presentation and tertiary care of cervical cancer patients in Uganda. We demonstrated that most patients presented with advanced stage cervical cancer. At UCI the national treatment targets for surgery and palliative care were met, however, the target for chemoradiotherapy was not met, mainly due to unavailability and limited capacity of radiotherapy facilities during the study period.

Our findings underestimate the care provision nationwide, since less than 10% of the cervical cancer patients reached UCI and more than half of the patients never initiated treatment or interrupted treatment. Our study highlights the importance of prevention including nationwide cervical cancer screening with a screen-and-treat approach, alongside effective referral pathways and investment in oncological surgical capacity and infrastructure, chemotherapy, and radiotherapy to lower the incidence of advanced stage disease and decrease the burden of cervical cancer in Uganda.

## Acknowledgments

We want to express our gratitude to all participating staff of the Uganda Cancer Institute for their support in conducting this study and retrieving the relevant medical records.

## Author Contributions

**Conceptualization:** Marlieke de Fouw, Carolyn Nakisige, Jogchum J. Beltman.

**Data curation:** Marlieke de Fouw, Melissa W. M. Boere.

**Formal analysis:** Marlieke de Fouw, Melissa W. M. Boere.

**Methodology:** Marlieke de Fouw, Jogchum J. Beltman.

**Supervision:** Carolyn Nakisige, Jogchum J. Beltman.

**Writing – original draft:** Marlieke de Fouw.

**Writing – review & editing:** Marlieke de Fouw, Melissa W. M. Boere, Carolyn Nakisige, Mariam Nabwire, Jane Namugga, Israel Luutu, Jackson Orem, Jan M. M. van Lith, Jogchum J. Beltman.

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
