## [Decision Letter · Decision Letter 0]

9 Oct 2024

PONE-D-24-14686Cervical cancer care at a tertiary oncology facility in Uganda: comparing daily practice with national treatment targets on cervical cancer controlPLOS ONE

Dear Dr. de Fouw,

Thank you for submitting your manuscript to PLOS ONE. After careful consideration, we feel that it has merit but does not fully meet PLOS ONE’s publication criteria as it currently stands. Therefore, we invite you to submit a revised version of the manuscript that addresses the points raised during the review process.

Kind regards,

Tim Luckett PhD

Academic Editor

PLOS ONE

Journal Requirements: When submitting your revision, we need you to address these additional requirements. 1. Please ensure that your manuscript meets PLOS ONE's style requirements, including those for file naming. The PLOS ONE style templates can be found at https://journals.plos.org/plosone/s/file?id=wjVg/PLOSOne_formatting_sample_main_body.pdf and https://journals.plos.org/plosone/s/file?id=ba62/PLOSOne_formatting_sample_title_authors_affiliations.pdf 2. We noted in your submission details that a portion of your manuscript may have been presented or published elsewhere. "We presented a summary of our findings at the EUROGIN conference in Bilbao in 2023" Please clarify whether this conference proceeding or publication was peer-reviewed and formally published. If this work was previously peer-reviewed and published, in the cover letter please provide the reason that this work does not constitute dual publication and should be included in the current manuscript. 3. Please review your reference list to ensure that it is complete and correct. If you have cited papers that have been retracted, please include the rationale for doing so in the manuscript text, or remove these references and replace them with relevant current references. Any changes to the reference list should be mentioned in the rebuttal letter that accompanies your revised manuscript. If you need to cite a retracted article, indicate the article’s retracted status in the References list and also include a citation and full reference for the retraction notice.

Reviewers' comments:

Reviewer's Responses to Questions

**Comments to the Author**

1. Is the manuscript technically sound, and do the data support the conclusions?

Reviewer #1: Yes

Reviewer #2: Yes

2. Has the statistical analysis been performed appropriately and rigorously? 

Reviewer #1: No

Reviewer #2: I Don't Know

3. Have the authors made all data underlying the findings in their manuscript fully available?

Reviewer #1: Yes

Reviewer #2: Yes

4. Is the manuscript presented in an intelligible fashion and written in standard English?

Reviewer #1: Yes

Reviewer #2: Yes

5. Review Comments to the Author

Reviewer #1: 

This study evaluated the clinical presentation and management of cervical cancer patients presenting at the Uganda Cancer Institute (UCI) in Kampala, the tertiary oncology facility in Uganda with access to radiotherapy. It is well-written and a very good addition to the oncology research from Africa.

Abstract: Well-written and concise abstract. Kindly strengthen the methods section by highlighting how data was analysed.

Introduction: Succinctly written introduction. Please ensure that the aim in the abstract is the same as the aim in the introduction section.

Methods:

Well-written methods section, with very good structure. It was easy to read. All relevant information was provided. However, please avoid the use of first-person pronouns such as “we”.

Please strengthen your rationale for selecting the timeframe “January 2017 and March 2018”. Kindly explain what makes this data still relevant for publication in 2024 since a lot has changed, including staging systems.

The analysis performed was very descriptive. Kindly consider performing inferential statistics to identify the determinants of the primary and secondary outcomes, including early or advanced stage presentation, diagnostic work-up, clinical management, national treatment targets and follow-up.

Results:

Good presentation of the results; however, very descriptive in nature.

Discussion

Good discussion. However, you can strengthen some of your two-sentence paragraphs. For example, “As reported in other studies from LMICs, the vast majority of women presented with advanced stage of disease (6, 7, 17, 22). This correlates with our finding that almost all patients had symptoms at time of presentation.”

Good for presenting the limitations of the study. Can you please highlight the strengths of the study as well?

Reviewer #2:

I greatly enjoyed this article and learned a lot about cervical cancer care in Uganda. I think the article would benefit from a clean copy edit. I noticed several instances in the text where there were discrepancies around comma usage, dangling participles, etc. Aside from a copy editing polish, I think the article would benefit from a little more of an explanation about how the methods utilized and data presented give us a clear understanding of how cervical cancer care access changed before and after the opening of the radiotherapy machine. Likewise, the conclusions of the article could use a little more work--what have we learned from this study about the ways in which radiotherapy access has and has not changed clinical care for cervical cancer patients? What does this study tell us about Uganda guidelines, etc. I think a read over of the Discussion section and ensuring that the authors guide us on how to make sense of the evidence presented would be helpful.

6. PLOS authors have the option to publish the peer review history of their article (what does this mean?). If published, this will include your full peer review and any attached files.

Reviewer #1: No

Reviewer #2: No

---

## [Author Response · Author response to Decision Letter 0]

26 Nov 2024

We would like to thank the reviewers for their thorough examination of our manuscript and helpful

comments. The quality of the manuscript has clearly benefited from the adjustments made following the comments. Please find below our responses to individual comments made by the reviewers and the feedback on the journal requirements. Please also find enclosed our revised manuscript with and without track changes.

1. Please ensure that your manuscript meets PLOS ONE’s style requirements, including those for file naming. 

Reply to journal requirement 1: 

We apologize for the inconvenience caused by not meeting PLOS ONE’s style requirements, and have adjusted our manuscript and file naming accordingly.

2. We noted in your submission details that a portion of your manuscript may have been presented or published elsewhere. "We presented a summary of our findings at the EUROGIN conference in Bilbao in 2023” 

Please clarify whether this conference proceeding or publication was peer-reviewed and formally published. If this work was previously peer-reviewed and published, in the cover letter please provide the reason that this work does not constitute dual publication and should be included in the current manuscript.

Reply to journal requirement 2:

The presentation during the EUROGIN conference in Bilbao was not peer-reviewed nor formally published. It was an oral presentation of our main findings as part of ‘Free communications’ in the session ‘Low income countries’. 

3. Please review your reference list to ensure that it is complete and correct. If you have cited papers that have been retracted, please include the rationale for doing so in the manuscript text, or remove these references and replace them with relevant current references. Any changes to the reference list should be mentioned in the rebuttal letter that accompanies your revised manuscript. If you need to cite a retracted article, indicate the article’s retracted status in the References list and also include a citation and full reference for the retraction notice

Reply to journal requirement 3:

We have reviewed our reference list and ensure it is complete and correct. 

Reviewer 1 comments:

Thank you for taking your time to read and review our manuscript. Below we have listed your comments and our feedback on a point-by-point basis.

1. Abstract: Well-written and concise abstract. Kindly strengthen the methods section by highlighting how data was analysed.

Reply to reviewer 1 comment 1:

Thank you for pointing out that we missed to include data analysis. We have adjusted the methods section as follows; 

We retrospectively analyzed medical records of all cervical cancer patients presenting to UCI between January 2017 and March 2018 for sociodemographic characteristics and clinical variables with descriptive statistics.

2. Introduction: Succinctly written introduction. Please ensure that the aim in the abstract is the same as the aim in the introduction section.

Reply to reviewer 1 comment 2:

Thank you for highlighting this omission in the abstract. We have corrected this in the revised manuscript as follows:

..This study evaluated the clinical presentation and management of cervical cancer patients presenting at the Uganda Cancer Institute (UCI) in Kampala, the tertiary oncology facility in Uganda with access to radiotherapy and reflected on daily clinical practice to identify priority areas for improving cervical cancer care in Uganda.

3. Methods:

Well-written methods section, with very good structure. It was easy to read. All relevant information was provided. However, please avoid the use of first-person pronouns such as “we”.

Please strengthen your rationale for selecting the timeframe “January 2017 and March 2018”. Kindly explain what makes this data still relevant for publication in 2024 since a lot has changed, including staging systems.

The analysis performed was very descriptive. Kindly consider performing inferential statistics to identify the determinants of the primary and secondary outcomes, including early or advanced stage presentation, diagnostic work-up, clinical management, national treatment targets and follow-up.

Reply to reviewer 1 comment 3:

Thank you for your positive review of our methods section. Considering your suggestion, we have adjusted the use of first-person pronouns and made changes accordingly to the methods section, highlighted with track changes.

Secondly, the timeframe between January 2017 and March 2018 was selected in order to demonstrate the effect of the unavailability of radiotherapy in Kampala on the treatment of cervical cancer patients, and to review how re-introduction of radiotherapy would alter the availability of treatment, since the new cobalt machine was introduced by the end of 2017. 

Thirdly, we acknowledge that since our data collection the situation has changed including the staging system for cervical cancer. Even though at the moment the access to radiotherapy in Kampala has increased, capacity is still limited, and we believe our study demonstrates it is valuable to compare the daily clinical practice with the ambitions formulated in national policies. We identified substantial gaps, its evaluation will support both clinicians and policy makers to identify priority areas considering the limited financial and human resources available. 

Other changes include recategorization of 2009 FIGO stage IB into three size range criteria and inclusion of nodal disease as a new stage IIIC. In addition, stage IA tumors are now defined only by depth of disease. We believe the change in staging system does not affect our outcomes much because unfortunately MRI is still not available to all cervical cancer patients to improve the accuracy of staging, and most patients will still present in advanced stage and need chemotherapy and radiation. Additionally, we have stratified FIGO 2009 stages in early and advanced stage which would not have changed the outcome when applying FIGO 2018 staging system. 

Lastly, we agree with your remark that the analysis is very descriptive in nature. We discussed this at length also with a statistician and concluded that descriptive statistics best illustrate our findings, since we aimed to describe the actual situation in clinical practice rather than comparing patients with early and advanced stage, and both patients groups (early and advanced stage) require different management with different treatment targets formulated in the national policy. Considering your recommendation, we have included additional analysis of the baseline characteristics of the groups, to better identify the differences between the patients presenting in early and advanced stage of disease, see table 1 in the revised manuscript and we added the following in the methods section:

The Chi-square test of independence was used for categorical sociodemographic and clinical variables between patients with early and advanced stage of cervical cancer, and the independent t-test for continuous variables. Statistical significance was set at p-value <0.05. 

4. Good presentation of the results; however, very descriptive in nature.

Reply to reviewer 1 comment 4:

For this comment we refer to our feedback provided on point 3 about the Methods section.

5. Discussion

Good discussion. However, you can strengthen some of your two-sentence paragraphs. For example, “As reported in other studies from LMICs, the vast majority of women presented with advanced stage of disease (6, 7, 17, 22). This correlates with our finding that almost all patients had symptoms at time of presentation.”

Good for presenting the limitations of the study. Can you please highlight the strengths of the study as well?

Reply to reviewer 1 comment 5:

Thank you for your positive feedback. We have elaborated a bit more on the topics presented, especially the HIV prevalence and relation to cancer prevalence, in order to strengthen the message and at the same time be concise and mindful of the length of the manuscript (see track changes in the manuscript) 

Following your interesting remark that our limitations came across more clearly than the strengths of the study, we have adjusted the title of the paragraph to ‘Strengths and limitations’ and added the following in the revised manuscript:

’Strengths and Limitations’ 

The present study provides a comprehensive overview of the presentation and actual management of cervical cancer patients at UCI, and is one of the few studies reporting on tertiary cervical cancer care in LMICs. Moreover, the study captures the impact of access to radiotherapy for cervical cancer patients, which is severely limited in LMICs and has devastating clinical implications.

Including all cervical patients presenting over a period of more than a year mitigates selection bias and therefore can be considered as a representative presentation of daily clinical practice. On the other hand, for 12% of patients (n=81), medical records could not be retrieved. The missing files were from patients presenting in different months, from different age groups and different districts, therefore selection bias is unlikely. 

Reviewer 2 comments:

Thank you for taking your time to read and review our manuscript. We are glad to hear that you have enjoyed reading our manuscript.

1. I think the article would benefit from a clean copy edit. I noticed several instances in the text where there were discrepancies around comma usage, dangling participles, etc.

Reply to reviewer 2 comment 1:

We have checked our manuscript for discrepancies and corrected them accordingly with track changes. Since the adjustments did not alter the content we do not specify them here. 

An independent reader has double checked our clean copy edit. 

2. Aside from a copy editing polish, I think the article would benefit from a little more of an explanation about how the methods utilized and data presented give us a clear understanding of how cervical cancer care access changed before and after the opening of the radiotherapy machine.

Reply to reviewer 2 comment 2:

Thank you for your comment to clarify this important issue with clinical implications further. In the paragraph ‘National treatment targets’ we describe that 19% of patients with an indication for chemotherapy and radiation received this treatment in the period when the cobalt machine broke down. After the cobalt machine was installed this proportion increased to 45% indicating that it has clearly increased access to the recommended care, however, the majority still did not access radiotherapy while the majority of the study cohort was in need of this treatment. 

3. Likewise, the conclusions of the article could use a little more work--what have we learned from this study about the ways in which radiotherapy access has and has not changed clinical care for cervical cancer patients? What does this study tell us about Uganda guidelines, etc. I think a read over of the Discussion section and ensuring that the authors guide us on how to make sense of the evidence presented would be helpful.

Reply to reviewer 2 comment 3:

Considering your remarks about radiotherapy access and clinical care in both this and the above comment, we have adjusted the discussion section with a specific focus on the impact of the radiotherapy availability. We refer to the track changes in the revised manuscript, since the length of the discussion is quite long to insert here.

---

## [Decision Letter · Decision Letter 1]

11 Dec 2024

Cervical cancer care at a tertiary oncology facility in Uganda: comparing daily practice with national treatment targets on cervical cancer control

PONE-D-24-14686R1

Dear Dr. de Fouw,

We’re pleased to inform you that your manuscript has been judged scientifically suitable for publication and will be formally accepted for publication once it meets all outstanding technical requirements.

Kind regards,

Tim Luckett

Academic Editor

PLOS ONE

**Reviewers' comments:**

Reviewer's Responses to Questions

1. If the authors have adequately addressed your comments raised in a previous round of review and you feel that this manuscript is now acceptable for publication, you may indicate that here to bypass the “Comments to the Author” section, enter your conflict of interest statement in the “Confidential to Editor” section, and submit your "Accept" recommendation.

Reviewer #1: All comments have been addressed

Reviewer #2: All comments have been addressed

2. Is the manuscript technically sound, and do the data support the conclusions?

Reviewer #1: Yes

Reviewer #2: Yes

3. Has the statistical analysis been performed appropriately and rigorously? 

Reviewer #1: Yes

Reviewer #2: N/A

4. Have the authors made all data underlying the findings in their manuscript fully available?

Reviewer #1: Yes

Reviewer #2: Yes

5. Is the manuscript presented in an intelligible fashion and written in standard English?

Reviewer #1: Yes

Reviewer #2: Yes

6. Review Comments to the Author

Reviewer #1: All my comments have been addressed by the authors.

No extra comments from me. Well done for strengthening your manuscript.

Reviewer #2: (No Response)

7. PLOS authors have the option to publish the peer review history of their article (what does this mean?). If published, this will include your full peer review and any attached files.

Reviewer #1: **Yes: **Andrew Donkor

Reviewer #2: No

---

## [Editor Report · Acceptance letter]

30 Dec 2024

PONE-D-24-14686R1 

PLOS ONE

Dear Dr. de Fouw, 

I'm pleased to inform you that your manuscript has been deemed suitable for publication in PLOS ONE. Congratulations! Your manuscript is now being handed over to our production team.

Kind regards, 

on behalf of

Dr. Tim Luckett 

Academic Editor

PLOS ONE